# DBE-Net: Dual Boundary-Guided Attention Exploration Network for Polyp Segmentation

**DOI:** 10.3390/diagnostics13050896

**Published:** 2023-02-27

**Authors:** Haichao Ma, Chao Xu, Chao Nie, Jubao Han, Yingjie Li, Chuanxu Liu

**Affiliations:** 1School of Integrated Circuits, Anhui University, Hefei 230601, China; 2Anhui Engineering Laboratory of Agro-Ecological Big Data, Hefei 230601, China

**Keywords:** polyp segmentation, colorectal cancer, boundary exploration, medical image analysis, deep learning, colonoscopy

## Abstract

Automatic segmentation of polyps during colonoscopy can help doctors accurately find the polyp area and remove abnormal tissues in time to reduce the possibility of polyps transforming into cancer. However, the current polyp segmentation research still has the following problems: blurry polyp boundaries, multi-scale adaptability of polyps, and close resemblances between polyps and nearby normal tissues. To tackle these issues, this paper proposes a dual boundary-guided attention exploration network (DBE-Net) for polyp segmentation. Firstly, we propose a dual boundary-guided attention exploration module to solve the boundary-blurring problem. This module uses a coarse-to-fine strategy to progressively approximate the real polyp boundary. Secondly, a multi-scale context aggregation enhancement module is introduced to accommodate the multi-scale variation of polyps. Finally, we propose a low-level detail enhancement module, which can extract more low-level details and promote the performance of the overall network. Extensive experiments on five polyp segmentation benchmark datasets show that our method achieves superior performance and stronger generalization ability than state-of-the-art methods. Especially for CVC-ColonDB and ETIS, two challenging datasets among the five datasets, our method achieves excellent results of 82.4% and 80.6% in terms of mDice (mean dice similarity coefficient) and improves by 5.1% and 5.9% compared to the state-of-the-art methods.

## 1. Introduction

Colorectal cancer (CRC) is a common digestive tract malignant tumor, which seriously endangers human health, and its incidence ranks third among all cancers [1]. As one of the most important precursors of colorectal cancer, polyps can easily transform into malignant tumors if not treated in time. A colonoscopy is an effective method to detect colon lesions, which can provide doctors with precise positioning information so that doctors can remove them in time before they become cancerous. However, colonoscopy occasionally overlooks cancer-causing polyps due to the inexperience of physicians. Therefore, automatic and accurate segmentation of polyps in colonoscopy images is of great significance for the clinical prevention of colorectal cancer.

At present, the existing methods of polyp segmentation can be divided into two categories, one is traditional methods, and the other is based on deep learning. Traditional polyp segmentation methods usually rely on handcrafted feature-based [2,3,4,5,6,7] methods to identify polyps, such as texture analysis, color distribution, geometric features, and intensity distribution. Though traditional methods have made considerable progress, these handcrafted feature-based methods still have low polyp segmentation accuracy and poor generalization ability, which cannot meet the requirements of clinical practice.

In recent years, with the continuous development of deep learning, many deep learning-based polyp segmentation methods [8,9,10,11,12,13,14,15,16,17,18,19] have been proven to outperform traditional handcrafted feature-based methods. With the proposal of the fully convolutional network FCN, the problem of image segmentation is introduced to the pixel level. Brandao et al. [20] and Vazquez et al. [21] used FCN to segment polyps from colonoscopy images, which achieved better segmentation accuracy than traditional methods. Subsequently, Ronneberger et al. [22] proposed a U-shaped network for biomedical image segmentation, namely U-Net, which is the most widely used network in medical image segmentation. Currently, U-Net++ [23], ResUnet++ [24], and other variant models based on the U-Net network architecture are applied to polyp segmentation, and this encoding–decoding network has become the mainstream network architecture in the medical field.

Recently, attention mechanisms have been increasingly applied to medical segmentation, especially polyp segmentation, which is often used to enhance the blurred boundaries of polyps [8,9,10] or to extract global and local features [25,26,27,28]. PraNet [8] uses an inverse attention mechanism to mine boundary information with a global feature map, which is generated by a parallel decoder that aggregates high-level feature information together. ACSNet [16] designed a spatial attention scheme to pay more attention to uncertain regions, but it did not explicitly model the relationship between ambiguous regions and boundary regions. Guo et al. [29] designed a ternary guidance module. This module suppresses background regions through a ternary guidance mask, thereby mining more polyp regions in uncertain regions without interference from tissues in the background. However, due to its limited ability to explore a wide range of uncertain regions, it does not fully utilize boundary information, and it is difficult to detect fine boundaries. Lai et al. [30] proposed a boundary-guided attention module that can guide the network to adaptively learn multi-scale boundary features for accurate polyp segmentation. However, it cannot explore more detailed information in uncertain regions, resulting in polyp segment segmentation and false positive segmentation. Inspired by the above, we propose a dual bounding-guided attention exploration module DBE, which can not only explore more uncertain regions in a wider range but also explore more detailed information with boundary masks.

For the multi-scale feature adaptation of polyps, there are some related studies dealing with the variation of polyp size and shape. MCNet [31] proposes a multi-scale context-guided deep network for automatic lesion segmentation with endoscopy images of the gastrointestinal tract, where both global and local contexts are captured as guidance for model training. MSNet [15] proposed a multi-scale subtraction network, which uses a subtraction unit (SU) to generate differential features between adjacent layers of the network and equips perceptual units at different levels with different perceptual fields in a feature pyramid manner, thus obtaining richer multi-scale difference information. Sun et al. [32] applied a dilated convolution at the end of the encoder to expand the receptive field to improve the performance of polyps, but it is difficult to capture more contextual information with only a dilated convolution. In view of the above, in our network, we propose a multi-scale context aggregation enhancement module CAM, which first aggregates the context information of different scales, and then adopts the channel feature pyramid to obtain the most extensive features from different receptive regions through multiple dilated convolutions with different dilation rates.

The transformer was originally proposed for the field of natural language processing (NLP). Vaswani et al. [33] first proposed a transformer for machine translation tasks. With the great success of transformers in the NLP field, Dosovitskiy et al. [34] proposed a vision transformer (ViT), which extended the transformer model architecture to the field of computer vision and was the first pure transformer for image classification. ViT can well replace the convolution operation and can still achieve good results in image classification tasks without relying on convolution. Chen et al. proposed a segmentation network TransUnet, that combines CNN and Transformer. TransUnet not only encodes image features as sequences through the transformer to enhance global context but also makes good use of low-level CNN features through a U-shaped hybrid structure design and sets a new record in CT multi-organ segmentation task. Zhang et al. [25] adopted a two-branch parallel architecture TransFuse, which runs a CNN-based encoder and a Transformer-based segmentation network in a parallel manner. Then, the multi-level features of the two branches are efficiently fused by using the proposed fusion module. Wang et al. [26] proposed a simple CNN-free backbone pyramid vision transformer (PVT) for dense prediction tasks. Compared with the design of ViT for image classification, PVT introduces the pyramid structure into the transformer so that it can be applied to various downstream dense prediction tasks. Compared with CNN, PVT performs better in detection and segmentation tasks. This paper adopts PVT as our encoder and designs a new polyp segmentation framework based on it.

In summary, the current polyp segmentation methods based on deep learning have greatly improved the segmentation accuracy and generalization ability compared with traditional methods, but these methods still have shortcomings when facing the challenge of polyp segmentation. In Figure 1, we provide several representative contrast images as an illustration. The first is the problem of blurred polyp boundaries, as shown in Figure 1a. When the colonoscope moves in the intestine, it will cause problems such as motion blur and reflections, which cause blurred boundaries of polyp images and increase the difficulty of polyp segmentation. The second is the multi-scale adaptability of polyps, as shown in Figure 1a–c. Due to the variable size and shape of polyp tissue, the current polyp segmentation methods still have certain limitations in the ability to extract multi-scale features. The third is the close resemblances between polyps and nearby normal tissues, as shown in Figure 1b,c. Polyps have low contrast to the background and high similarity in texture and color to surrounding tissue, which makes accurate identification of polyp areas difficult.

To solve the above problems, this paper proposes a novel and efficient dual boundary-guided attention exploration network (DBE-Net) for accurate polyp segmentation. This network uses the cooperative exploration of uncertainty regions and border regions as the primary driving force for polyp segmentation. First, to deal with the problem of blurred polyp boundaries, we design a dual-boundary guided attention exploration module (DBE), which gradually approximates real polyp boundaries through a coarse-to-fine strategy. DBE is composed of a ternary mask submodule (TM) and a boundary mask submodule (BM). First, we use TM to suppress the background area, balance the foreground area and enhance the uncertainty area, which can explore more polyp areas in a wider uncertainty area and reduce the interference of normal tissues in the background. Second, we utilize BM to guide the network to adaptively learn the perceptual features of the boundary region to achieve re-enhancement of the polyp boundary. Meanwhile, to address the multi-scale feature adaptation problem, we propose a multi-scale context aggregation augmentation module (CAM). This module first aggregates the features of two adjacent layers to obtain the feature information of different levels. Then, the feature maps are convolved with dilated convolutions with different dilation rates through the channel feature pyramid to accommodate the variation in polyp size and shape. In addition, low-level features usually contain rich, detailed information such as texture, color, and edge information. To address the close resemblances between polyps and nearby normal tissues, we propose a low-level detail enhancement module (LDE). This module can extract more low-level details and provide them with more high-level features for fusion after up-sampling operation to promote the overall performance of the network. Meanwhile, this module uses the combination of channel attention and spatial attention to capture polyp details in different dimensions.

Our main contributions can be highlighted as follows:We propose a new dual boundary-guided attention exploration network termed DBE-Net. Different from existing CNN-based methods, this network adapts the pyramid vision transformer as the encoder to obtain more robust features and is combined with the partial decoder PD as the base framework.We propose a dual boundary-guided attention exploration module, termed DBE. This module adopts a coarse-to-fine strategy to gradually approximate the real polyp boundary from coarse uncertain regions to fine boundary regions through ternary masks and boundary masks according to the prediction results of previous scales to obtain a clearer polyp segmentation boundary.We propose a multi-scale context aggregation augmentation module termed CAM. To address the multi-scale feature adaptation problem, this module adopts multiple dilated convolutions with different dilation rates and obtains the most extensive features from different receptive regions to adapt to the multi-scale changes of polyps.To solve the problem of high similarity between polyps and surrounding tissues, we propose a low-level detail enhancement module, termed LDE. This module can extract more low-level details to improve the overall network performance and achieve more accurate polyp segmentation.

## 2. Materials and Methods

### 2.1. Proposed Network Structure

The purpose of polyp segmentation is to identify polyp regions from normal tissues in colonoscopy images and to segment polyps robustly. However, the polyp boundaries are irregular, even somewhat blurred, and disturbed by normal tissues in the background, which brings great challenges to improving polyp segmentation performance. Therefore, we propose a dual boundary-guided attention exploration network DBE-Net, which will be used for accurate polyp segmentation.

As shown in Figure 2, the proposed network mainly consists of four parts, including a pyramid vision transformer (PVT) encoder, dual boundary-guided attention exploration module (DBE), multi-scale context aggregation enhancement module (MCA), and low-level detail enhancement module (LDE). In the encoder stage, to obtain more powerful backbone features and provide more foreground information for subsequent decoding stages, PVTv2 is adopted as our encoder. Compared with traditional CNN methods, which pay more attention to local information, PVTv2 exhibits stronger global information extraction ability and better robustness to input disturbances. In the decoder stage, considering that the transformer is good at providing global information but lacks detailed local information, it makes it difficult to solve the multi-scale feature adaptation problem, including size and shape changes. To cope with this problem, MAC can obtain a wider range of features from different receptive regions by aggregating and enhancing the features of each stage to adapt to the multi-scale changes of polyps, thereby obtaining more abundant local and global features. Then, a partial decoder (PD) obtains preliminary rough segmentation results by aggregating features from the outputs of the MAC modules at different stages. However, the preliminary results obtained by the PD module can only obtain significant foreground regions, and it is difficult to judge accurate polyp boundaries for fuzzy boundaries. Therefore, the DBE adopts a coarse-to-fine strategy to segment polyps. According to the preliminary prediction results obtained by PD, the real polyp boundaries are gradually approximated from rough uncertain regions to fine boundary regions by the ternary mask sub-module and the boundary mask sub-module. Low-level features usually contain rich detail information, while the detail fusion sub-module (DF) in LDE can extract more low-level detail information and provide it to high-level features through up-sampling. At the same time, the detail extraction sub-module (DE) can capture the detailed appearance information to promote the overall performance of the network.

In our network model, we use PVTv2 as the backbone network to extract four multi-scale pyramid features ***X****_i_* from the input image ***I***, where *i* ∈ {1, 2, 3, 4}. Then, every two adjacent features of these four features are sent to MCA for multi-scale context aggregation enhancement, leading to three output features ***M***_2_, ***M***_3_, and ***M***_4_. Next, we feed the output features ***M***_2_, ***M***_3_, and ***M***_4_ to PD to fuse, resulting in preliminary rough segmentation results ***D***_5_. At the same time, the coarse segmentation result ***D***_5_ from PD and low-level features ***X***_1_ are fed to LDE for detail enhancement to obtain ***L****_i_*, where *I* ∈ {1, 2, 3, 4}. Then the PD features ***D***_5_ are fed into the DBE, which is combined with the detail enhancement features ***L***_4_ from the LDE output to obtain a clearer boundary segmentation through two stages of boundary exploration from coarse to fine. Meanwhile, the output features ***D***_4_ and ***T***_4_ of the two stages serve as side supervision. Then, the enhanced features ***D***_4_ are up-sampled and fed to the next DBE. There they are again combined with the detail enhancement features ***L***_3_ to further obtain a clearer boundary segmentation and output two-stage side supervision ***T***_3_ and ***D***_3_. This process is repeated in detail enhancement features ***L***_2_ and ***L***_1_ while obtaining four side supervisions, ***T***_2_, ***D***_2_, ***T***_1_, and ***D***_1_. Finally, through four stepwise boundary explorations, the final predicted output ***D***_1_ is obtained.

Our overall process is as follows:(1)$$X_{i}=T(I)(i=1,2,3,4)$$
(2)$$M_{i}=M(X_{i},X_{i-1})(i=2,3,4)$$
(3)$$D_{5}=P(M_{2},M_{3},M_{4})$$
(4)$$L_{i}=L(D_{5},X_{1},X_{2},X_{3},X_{4})(i=1,2,3,4)$$
(5)$$D_{1}=D(L_{1},D(L_{2},D(L_{3},D(L_{4},D_{5}))))$$

Here: (1) ***I*** represents the input image; (2) ***T***, ***M***, ***P***, ***L***, and ***D*** represent the functions of PVTv2, MAC, PD, LDE, and DBE, respectively.

### 2.2. Dual Boundary-Guided Attention Exploration Module

When the colonoscope moves in the intestine, it may produce problems such as motion blur and reflection, which causes blurred boundaries in the polyp image. To pay more attention to the pixels of the blurred boundary and achieve a clearer polyp segmentation boundary, we design a dual boundary-guided attention exploration module DBE, which adopts a coarse-to-fine gradual approximation strategy. First, according to the prediction results of the previous scale and detail enhancement features, more ground-truth polyp regions are obtained from rough uncertain regions through a ternary mask sub-module. Then, the boundary mask sub-module explores the finer polyp boundaries from the fine boundary regions. Finally, multiple DBE operations are used to gradually approach the true polyp boundary. Figure 3 shows the detailed structure of the DBE. This module consists of two sub-modules, namely the ternary mask sub-module TM and the boundary mask sub-module BM.

For the TM sub-module, it divides the feature map into three regions, namely foreground, background, and uncertainty regions. The foreground represents polyp regions with high response, the background represents non-polyp regions with low response, and uncertainty regions refer to fuzzy regions with intermediate responses. The uncertain area includes foreground and background objects and is a mixture of the two. The purpose of the TM sub-module is to mine as many foreground objects as possible from the uncertain region and exclude background objects. Specifically, the ternary mask assigns different weight values to different regions. To emphasize the uncertain area, the pixels in the uncertain area are set to the highest weight of 1. Pixels in the foreground area are set to 0 to balance the highly responsive areas. To suppress the interference of the background area, the pixels in the background area are set to the lowest weight of −1. The specific process is as follows:

As shown in the gray rounded rectangle box in Figure 3, the low-resolution output feature map ***D***_*i*+1_ by the previous stage is up-sampled to obtain ***D***_*u*_.
(6)$$D_{u}=U(D_{i+1})$$
where ***U*** represents the up-sampling operation.

Then, a ternary mask ***D***_*t*_ (Equation (7)) is generated based on two thresholds, **α*_l_*** and **α*_h_***. In Equation (7), the value of *i*-th pixel of ***D***_*u*_ is greater than **α*_h_***, which means it belongs to the foreground region and sets the weight to 0. If it is less than **α*_l_***, it belongs to the background area and sets the weight to −1. When it is between **α*_l_*** and **α*_h_***, it belongs to the uncertainty region and sets the weight to 1.
(7)$$D_{t}^{i}=\left\{ \begin{array}{cc} -1, & if\;D_{u}^{i}\leq \alpha_{l}\\ 1, & if\;\alpha_{l}\leq D_{u}^{i}\leq \alpha_{h}\\ 0, & if\;D_{u}^{i}\geq \alpha_{h} \end{array}\right.$$

Next, the generated ternary mask ***D***_*t*_ is element-wise multiplied by the features ***L****_i_* from the current stage to obtain weighted feature maps ***L****_t_*. Then, ***L****_t_* is 3 × 3 convoluted and combined with ***D***_*u*_ through element-wise addition to generating the output feature maps ***T****_i_* of the TM sub-module. In particular, the feature maps ***T**_i_* is used both as side supervision and as the input of the BM sub-module.
(8)$$T_{i}=f_{i}(L_{i}⊗D_{t})+D_{u}$$
where ⊗ represents element-wise multiplication. ***f****_i_* denotes 3 × 3 convolution.

For the BM sub-module, it can extract richer and more detailed boundary field information, which can help to explore more accurate polyp edges. To achieve the goal of enhancing boundary features, we extract boundary information from the output feature maps ***T****_i_* of the previous DM sub-module through morphological operations. As shown in the cyan rounded rectangular box in Figure 3, the output feature maps ***T****_i_* of the previous sub-module DM is converted into a binary mask ***D***_*s*_ with a threshold of 0.5. If the probability of a pixel belonging to a polyp is greater than 0.5, the pixel is classified as a foreground polyp pixel; otherwise, it is classified as a background pixel. Then, we use morphological operations on the binary mask ***D***_*s*_:(9)$$D_{m}=Dilate(D_{s},E)-Erode(D_{s},E)$$
where ***Dilate(D_s_***, ***E)*** denotes the morphological dilation operation on the binary mask ***D***_*s*_ using multi-structure elements ***E*** to generate a dilated mask ***D***_*d*_. ***Erode(D_s_, E)*** denotes the morphological erosion operation on the binary mask ***D***_*s*_ using multi-structure elements ***E*** to generate an erosion mask ***D***_*e*_. The morphological dilation operation dilates the polyp border outward, thereby expanding the foreground area. The morphological erosion operation shrinks the polyp boundary, thereby reducing the foreground area.

Similar to [36], we also consider multi-structure elements. Let (*m, n*) be the center of the binary mask ***D***_*s*_, and a structural element in the (2*N* + 1) × (2*N* + 1) square window can be represented as:(10)$$E_{i}=\left\{M(m+m_{0},n+n_{0}),\theta_{i}=i\times \alpha |-N\leq m_{0},n_{0}\leq N\right\}$$
where *i* = 0, 1, …, 4*N* − 1, *α* = 180°/4*N,* and *θ_i_* is the direction angle of the structure element.

In this paper, for the two DBE modules at higher levels, we choose *N* = 2, and then in a 5 × 5 square window, the direction angles of all structure elements are 0°, 22.5°, 45°, 67.5°, 90°, 112.5°, 135°, and 157.5°. For the two DBE modules at the lower level, we choose *N* = 3, and then in the 7 × 7 square window, the direction angles of all structure elements are 0°, 15°, 30°, 45°, 60°, 75°, 90°, 105°, 120°, 135°, 150°, and 165°. They contain most of the directions in which the lines extend in the image. In morphological dilation, each structural element ***E****_i_* in different directions was used to dilate the image, and the averaged result is taken as the dilated mask. Morphological erosion operation is similar. Then, we perform an element-wise subtraction operation on ***D***_*d*_ and ***D***_*e*_ to generate the boundary mask ***D***_*m*_.

Next, the generated ternary mask ***D***_*m*_ is element-wise multiplied by the input features ***L****_i_* from the previous TM sub-module to obtain boundary-enhanced feature maps ***L****_m_*. Then, ***L****_m_* is 3 × 3 convoluted and combined with ***T****_i_* through element-wise addition to generating the output feature maps ***D****_i_* of the DM sub-module. Meanwhile, ***D****_i_* also serves as the final output of the DBE module. In particular, the feature maps ***D****_i_* is used both as side supervision and as the input of the next DBE module.
(11)$$D_{i}=f_{i}(L_{i}⊗D_{m})+T_{i}$$

### 2.3. Multi-Scale Context Aggregation Enhancement Module

Due to the variable size and shape of polyp tissue, current polyp segmentation methods still have certain limitations in their ability to extract multi-scale features. To address the multi-scale feature adaptation problem of polyps, we introduce a multi-scale context aggregation augmentation module MCA, which can obtain more detailed local and global feature information, as shown in Figure 4. This module complements the location and spatial information through the interaction of adjacent high- and low-layer feature information. At the same time, this module uses the channel feature pyramid (CFP) [37] module to obtain the most extensive multi-scale feature information from different receptive regions to adapt to the multi-scale variation of polyps.

Specifically, we use 1 × 1 convolution on two feature maps from different layers to reduce feature channels to 32 to reduce computational resources. The features from lower layers are down-sampled and connected with the features from higher layers as the input of CFP. The CFP module has four branches, and the dilate rates of different branches are set to (1, 2, 4, 8) to obtain more contextual information. Each branch contains three groups of convolution blocks. Each group of convolution blocks decomposes the standard convolution into 3 × 1 convolution and then performs 1 × 3 convolution. Then, with this factorized convolution [37], 33% of the parameters can be saved with the same number of filters. Finally, the features of the four branches are concatenated together as the output of the MCA module.

### 2.4. Low-Level Detail Enhancement Module

The appearance of polyps is usually very similar to the surrounding normal tissue; however, low-level feature maps usually contain more detailed information, such as texture, borders, and color. Therefore, we design a low-level detail enhancement module LDE to extract richer detailed information. As shown in Figure 5, the LDE module consists of two parts, namely the detail fusion sub-module DF and the detail extraction sub-module DE.

First, the DF sub-module reduces the interference of background information by the preliminary segmentation result ***D***_5_ from the PD module. Then, more detailed information is provided on high-level features for fusion through down-sampling to improve the performance of the overall network and improve the accuracy of polyp segmentation. As shown in the grey rounded rectangle box in Figure 5, the features ***D***_5_ from the PD module are transformed into a binary mask ***M*** with a threshold α. If the probability of belonging to a polyp pixel is less than α, it is classified as a background pixel with a weight of 0. Where α is set to 0.2.
(12)$$M^{i}=\left\{ \begin{array}{cc} 0, & if\;X_{1}^{i}\leq \alpha \\ 1, & if\;X_{1}^{i}\geq \alpha \end{array}\right.$$

Next, the generated mask ***M*** is element-wise multiplied by the low-level features ***X***_1_ to obtain feature maps ${X_{1}}^{\prime }$ with reduced background noise. Then, ${X_{1}}^{\prime }$ is up-sampled and concatenated with ***X***_2_, ***X***_3_, and ***X***_4,_ respectively, to generate detail enhancement feature maps ***L***_2_, ***L***_3_, and ***L***_4_.
(13)$${X_{1}}^{\prime }=X_{1}⊗M$$

For the DE sub-module, we mainly adopt the tandem use of channel attention operation ***A***_*c*_(·) and spatial attention operation ***A***_*s*_(·) to capture detailed information on polyps from different dimensions.
(14)$$L_{1}=A_{s}(A_{c}(X_{1}))$$

The channel attention module formula is as follows:(15)$$A_{c}(F)=\sigma (MLP(P_{avg}(F))+MLP(P_{max}(F)))⊗F$$
where ***P****_avg_* and ***P****_max_* represent the max pooling and average pooling functions, respectively. For the input feature map *F*, the ***P****_avg_* and ***P****_max_* functions are first applied at each spatial location, resulting in two C × 1 × 1 feature maps. Then each is fed into a shared *MLP* containing two fully connected layers. Finally, the channel attention map is obtained by adding pixels and passing through an activation function.

The spatial attention module formula is as follows:(16)$$A_{s}(F)=\sigma (f^{7\times 7}(Concat(P_{avg}(F),P_{max}(F))))⊗F$$
where the feature maps output by the channel attention operation is taken as the input of the spatial attention operation. First, the ***P****_avg_* and ***P****_max_* functions are applied at each channel, resulting in two C × 1 × 1 feature maps. Then, they are connected by channel and fed into a 7 × 7 convolutional layer. Finally, the spatial attention map is obtained through the activation function.

### 2.5. Loss Function

The DBE-Net is trained via a deep supervision manner. There are a total of 9 aspects of supervision across all stages of the network (Figure 1), including ***D****_i_* (*i =* 1, 2, 3, 4, 5) and ***T****_i_* (*i =* 1, 2, 3, 4, 5). In the experiments, we use the weighted binary cross entropy (***L***_*wbce*_) loss and the weighted intersection over union (***L***_*wiou*_) loss to supervise each ***D****_i_* and ***T****_i_*. These two losses restrict the prediction map from the global object perspective and local pixel perspective, respectively. The total loss can be expressed as:(17)$$L=L_{main}+L_{aux}$$
where ***L***_*main*_ and ***L***_*aux*_ represent the main loss and auxiliary loss, respectively. The main loss ***L***_*main*_ is to calculate the loss between the feature map ***D****_i_* and the ground truth ***G***, which can be expressed as follows:(18)$$L_{main}=\sum_{i=1}^{5}\left(L_{wbce}(D_{i},G)+L_{wiou}(D_{i},G)\right)$$

The auxiliary loss ***L***_*aux*_ is to calculate the loss between the feature map ***T****_i_* and the ground truth ***G***, which can be expressed as follows:(19)$$L_{aux}=\sum_{i=1}^{4}\left(L_{wbce}(T_{i},G)+L_{wiou}(T_{i},G)\right)$$

### 2.6. Datasets

To evaluate the segmentation performance of our proposed method, we conduct experiments on five polyp segmentation data, including Kvasir-SEG [38], CVC-ClinicDB [39], CVC-ColonDB [35], ETIS [21] and CVC-T [5]. The description of these datasets is as follows.

(1)**Kvasir-SEG**. The dataset consists of 1000 polyp images and their corresponding ground truth polyp masks annotated by endoscopy experts. The resolution of the images contained in Kvasir-SEG varies from 332 × 487 to 1920 × 1072 pixels.(2)**CVC-ClinicDB**. The dataset, also known as *CVC-612*, contains 612 open-access images from 25 colonoscopy videos with a resolution of 384 × 288.(3)**CVC-ColonDB**. The dataset consists of 380 images from 15 colonoscopy videos with an image resolution of 574 × 500.(4)**ETIS**. The dataset contains 196 images captured from 34 colonoscopy videos with an image resolution of 1225 × 996.(5)**CVC-T**. This dataset is a subset of EndoScene and contains 60 images from 44 colonoscopy sequences of 36 patients with an image resolution of 574 × 500.

For a fair comparison, we use the same training and test dataset splits as [8,17,18], as shown in Table 1. A total of 1450 training images are used in the training set, of which 900 images are from Kvasir-SEG, and 550 images are from CVC-ClinicDB. A total of 798 test images are used in the test set, of which 100 images are from Kvasir-SEG, and 62 images are from CVC-ClinicDB. Meanwhile, the other three datasets are used for testing, including 380, 196, and 62 images in CVC-ColonBD, ETIS, and CVC-T, respectively.

## 3. Results

In this section, we employ three kinds of experiments to validate the performance of DBENet, including qualitative and quantitative experiments and an ablation study. In the experiments, we compare our method with six different methods, including state-of-the-art methods, including UNet [22], UNet++ [23], SFA [40], PraNet [8], SANet [17] and CaraNet [18]. Meanwhile, we analyze and compare three evaluation metrics on five different polyp segmentation datasets. For a fair comparison, the segmentation maps for these methods are published by the authors or directly generated from the original code provided by the authors.

### 3.1. Training Settings and Evaluation Metrics

Our proposed model is implemented in Pytorch with NVIDIA RTX3090 for GPU acceleration. Before feeding into the network, we resize the input image to 352 × 352. Instead of data augmentation, we use a multi-scale training strategy {0.75, 1.0, 1.25}. We start the training with a batch size of 16 and adopt the Adam optimization method to optimize the model parameters. The initial learning rate is set to 0.0001, and the maximum epoch is set to 100. Thresholds **α*_l_*** and **α*_h_*** are set to 0.3 and 0.7, respectively.

In this paper, the mean dice similarity coefficient (*mDice*), the mean intersection over union (*mIoU*), and the mean absolute error (*MAE*) were used to quantitatively evaluate the segmentation performance of polyps. For these evaluation metrics, higher values of mDice and *mIoU* indicate better segmentation results, while lower values of *MAE* indicate better segmentation results.

The Dice coefficient is an ensemble similarity measure and a standard measure for comparing pixel-level classification results between predicted masks and ground truth values. The *mDice* represents the average of the dice similarity coefficients between all predicted images and the ground truth, and the formula is as follows.
(20)$$mDice=\frac{1}{n+1}\sum_{i=1}^{n}\frac{2TP}{FP+2TP+FN}$$
where *TP*, *FP*, *FN*, and *n* denote true positive, false positive, and false negative predictions and the number of test images, respectively.

The IoU calculates the overlap ratio between the predicted mask and its corresponding ground truth value. The *mIoU* represents the average of the IoU values between all predicted images and the ground truth, and the formula is as follows.
(21)$$mIoU=\frac{1}{n+1}\sum_{i=1}^{n}\frac{TP}{FP+TP+FN}$$

*MAE* is used to evaluate pixel-level accuracy. We evaluate pixel-level accuracy by computing the mean absolute error between the model’s predicted and true values, and the formula is as follows.
(22)$$MAE=\frac{1}{n}\sum_{i=1}^{n}\left|{\hat{p}}_{i}-p_{i}\right|$$
where ${\hat{p}}_{i}$ and $p_{i}$ denote the prediction and the corresponding ground truth for the *i*-th pixel out of the total *n* pixels.

### 3.2. Qualitative Analysis

Figure 6 shows a visual comparison between our model DBE-Net and different competing methods. Apparently, our method can segment polyp regions more accurately and outperform several challenging aspects compared to other competing methods. First, DBE-Net can obtain clearer polyp boundaries and reduce the interference of complex backgrounds such as intestinal folds and dim areas. For example, intestinal folds were mistakenly missed by other methods in Figure 6e, and polyps in dim regions were missed or misjudged by other methods in Figure 6a. In both cases, DBE-Net can accurately detect polyps, which may be attributed to the role of the DBE module. This module gradually enhances the fuzzy edge information through a coarse-to-fine strategy to achieve the purpose of approximating the real polyp boundary. Second, DBE-Net can effectively deal with the multi-scale changes of polyps and identify large polyps and small polyps of different shapes, especially to avoid misjudgment of small polyps. As shown in Figure 6a–c, only our method can accurately detect small polyps and reduce false positives. Third, DBE-Net can also better solve the problem of high similarity between polyps and surrounding tissues. As shown in Figure 6d,e, only our method can segment the polyp region more completely.

### 3.3. Quantitative Experiments

Table 2 and Table 3 present the quantitative results of the statistical comparison of DBE-Net with six different competing methods on five different datasets. For ease of viewing, the best results for each evaluation metric are highlighted in bold. Considering that the training set is selected from CVC-ClinicDB and Kvasir-SEG, the learning ability of the proposed method is first analyzed quantitatively. As shown in Table 2, our proposed method is optimal on various metrics on both datasets. The mDice of our method achieves excellent results of 93.9% and 92.2% in CVC-ClinicDB and Kvasir-SEG.

To verify the generalization abilities of the model, we test it on three unseen datasets (CVC-ColonDB, ETIS, and CVC-T). As shown in Table 3, the results of the proposed method are still optimal and achieve good generalization performance compared to the six competing methods. Especially, as shown in Figure 7, the generalization ability of our method achieves a significant improvement on the challenging datasets CVC-ColonDB and ETIS. On the CVC-ColonDB dataset, the mDice of our method outperforms CaraNet and SANet by 5.1% and 7.1%, respectively. In terms of the ETIS dataset, the mDice of our method outperforms CaraNet and SANet by 5.9% and 5.6%, respectively. On the CVC-T dataset, the mDice of our method surpasses SANet by 1.5%.

### 3.4. Ablation Studies

To verify the effectiveness of each component in the model, we conduct ablation experiments on the dual boundary-guided attention exploration module (DBE), multi-scale context aggregation enhancement module (MCA), and low-level detail enhancement module (LDE). The baseline of our backbone network is composed of PVTv2 and PD, and the standard model is composed of “Baseline + MCA + LDE + DBE”. We evaluate the effectiveness of different modules by removing or changing them from the standard model. “w/o MCA”, “w/o LDE,” and “w/o DBE” represent the removal of MCA, LDE, or DBE, respectively, from the standard model.

Ablation studies of DBE: To investigate the effectiveness of the DBE module, we first visualize the output characteristics of the DBE module at different stages. As shown in Figure 8, the redder the area, the more attention the network needs to give. We can observe that it gradually refines the polyp boundary and explores more polyp information as the stage progresses. This shows that the coarse-to-fine strategy of the DBE module can effectively enhance the fuzzy edge information. To quantitatively analyze the effectiveness of the DBE module, we train a version “w/o DBE” that lacks the DBE module. We removed the DBE module altogether and replaced it with an element-wise addition operation. The experimental results are shown in Table 4. Compared to the standard model, the performance of the model without DBE dropped sharply on the five datasets. As shown in the dumbbell diagram in Figure 9, on CVC-ClinicDB, Kvasir-SEG, CVC-ColonDB, ETIS, and CVC-T, compared to the performance of the standard model DBENet, the absence of DBE reduces mDice by 0.8%, 0.6%, 1.7%, 1.6%, and 2.1%, respectively.

Ablation studies of MCA: Similarly, to quantitatively analyze the effectiveness of the MCA module, we train a version “w/o MCA” that lacks the MCA module. We remove all MCA modules and directly fuse the context information of the three high layers (i.e., ***X***_2_, ***X***_3,_ and ***X***_4_). The results in Table 4 show that compared with the standard model, the model lacking MCA has a 1.0% and 0.5% drop in mDice on the CVC-ColonDB and ETIS datasets, respectively. In particular, the CVC-ColonDB and ETIS datasets are challenging unseen datasets, which suggests that MCA is helpful for multi-scale generalization.

Ablation studies of LDE: To demonstrate the power of LDE, we train a version “w/o LDE” that lacks the LDE module. As shown in Table 4, removing the LDE results in a slight drop in performance on all five datasets compared to the standard model. On the five datasets, CVC-ClinicDB, Kvasir-SEG, CVC-ColonDB, ETIS, and CVC-T, removing LDE can reduce mDice by 0.5%, 0.4%, 0.6%, 0.1%, and 1.0%, respectively. Therefore, the LDE module plays a role in improving the overall segmentation ability of the model.

## 4. Discussion

Colonoscopy is an effective way to detect colon lesions. Automatic segmentation of polyp regions can help doctors improve the accuracy of diagnosis. The current polyp segmentation research based on deep learning still has the following problems: blurry polyp boundaries, multi-scale adaptability of polyps, and close resemblances between polyps and nearby normal tissues. For the fuzzy polyp boundary problem, we use a coarse-to-fine strategy to gradually enhance the true polyp boundary. As can be seen from Figure 8, as the stage goes on, the polyp boundary becomes clearer, and the background interference is more suppressed, which shows that our strategy can effectively enhance the blurred edge information. For the multi-scale adaptability of polyps, the proposed MCA module first supplements position and spatial information by fusing adjacent high- and low-layer feature information and then uses dilated convolutions with different dilation rates to obtain the most extensive multi-scale feature information from different receptive areas to accommodate multi-scale changes in polyps. As shown in Figure 6, our model can effectively identify polyps of different shapes and sizes, especially for the misjudgment of small polyps. For the similarity between polyps and adjacent normal tissues, the proposed LDE module makes full use of the low-level features that usually contain more detailed information, such as texture, borders, and color. We reduce the interference of background information in the low-level features through the initial segmentation result *D*_5_ and then capture the detailed information of polyps from different dimensions through the tandem use of channel attention operation and spatial attention operation. From the results of the ablation experiments in Table 4, it can be concluded that the LDE module promotes the overall performance of the model.

The proposed DBE-Net model achieves superior results compared to state-of-the-art methods on all five public polyp segmentation benchmark datasets. From the qualitative results, as shown in Figure 6, it is obvious that the proposed model can obtain sharper segmentation masks while reducing the complex background interference. For polyps of different shapes and sizes, our model can still identify them effectively. Compared with CaraNet, the proposed model improved the mDice by 0.3% and 0.4% on CVC-ClinicDB and Kvasir-SEG datasets, respectively. Test results on three unseen datasets show that our model exhibits a stronger generalization ability. Especially for CVC-ColonDB and ETIS, two challenging datasets among the five datasets, our method achieves excellent results of 82.4% and 80.6% on mDice and improves by 5.1% and 5.9% compared with CaraNet, respectively. This result is significantly ahead of the current state-of-the-art methods.

In this paper, the presented study still has some limitations. We speculate that more data enhancement processing of the original dataset could further improve the efficiency of the model, including increasing the size of larger datasets, applying more image enhancement techniques, and image post-processing steps. The DBE-Net model allows us to design a very deep and clever neural network architecture to achieve more accurate polyp segmentation. However, DBE-Net should not only be limited to the polyp segmentation domain but also can be extended to other medical image segmentation, which will help us to further promote the generalization ability of the proposed model. Furthermore, the proposed model can also be extended to natural image segmentation and other pixel classification tasks. We used all our experience and knowledge to optimize the model, but there may be further optimizations, which may affect the performance of the method.

## 5. Conclusions

In this paper, we propose a new polyp segmentation framework, DBE-Net, to help address three major challenges in the field of polyp segmentation: blurry polyp boundaries, multi-scale adaptability of polyps, and close resemblances between polyps and nearby normal tissues. To address the blurred boundary problem, we introduce the dual boundary-guided attention exploration module DBE. This module mainly adopts a coarse-to-fine strategy to gradually approach the real polyp boundary. Meanwhile, the proposed network introduces a multi-scale context aggregation enhancement module MCA and a low-level detail enhancement module LDE to help address the problem of polyp multi-scale adaptability and the similarity of polyps to surrounding tissues. Extensive qualitative and quantitative experiments demonstrate that DBE-Net outperforms current state-of-the-art models. For performance analysis on unseen datasets, it is found that the proposed network has a higher generalization ability than other state-of-the-art methods. Especially for the unseen CVC-ColonDB dataset and ETIS dataset, the mDice of the proposed model reaches 82.4% and 80.6%. This result is significantly ahead of current state-of-the-art models. We can conclude that the proposed DBE-Net model can accurately segment polyps and can be considered a powerful polyp segmentation method. Further, we can develop a stable and clinically applicable system for polyp segmentation from colonoscopy video frames.

## Figures and Tables

**Figure 1 diagnostics-13-00896-f001:**
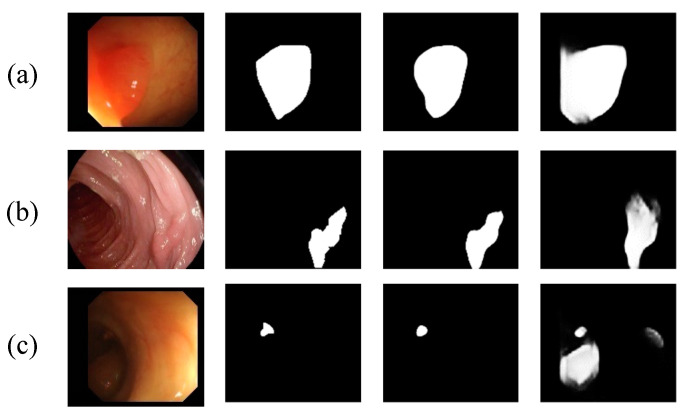
The segmentation examples of our model and CaraNet [18] with different challenge cases: (**a**) blurred boundaries of polyps, (**a**–**c**) multi-scale adaptability of polyps, (**b**,**c**) close resemblances between polyps and nearby normal tissues. These images are from unseen datasets CVC-ColonDB [35] and ETIS [21], which show that our model has better generalization ability.

**Figure 2 diagnostics-13-00896-f002:**
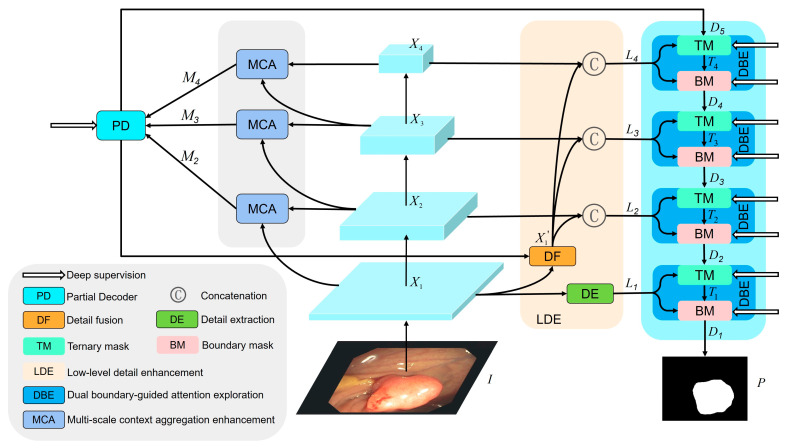
Overall architecture of the proposed DBE-Net, which consists of a pyramid vision transformer (PVT) encoder, dual boundary-guided attention exploration module (DBE), multi-scale context aggregation enhancement module (MCA), and low-level detail enhancement module (LDE).

**Figure 3 diagnostics-13-00896-f003:**
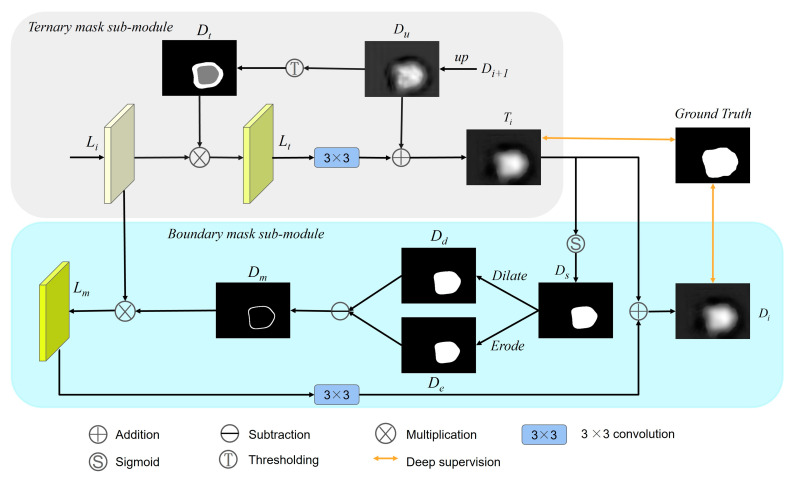
The structure of DBE, which consists of two sub-modules, including the ternary mask sub-module and the boundary mask sub-module. It adopts a coarse-to-fine strategy to gradually approximate the real polyp boundary.

**Figure 4 diagnostics-13-00896-f004:**
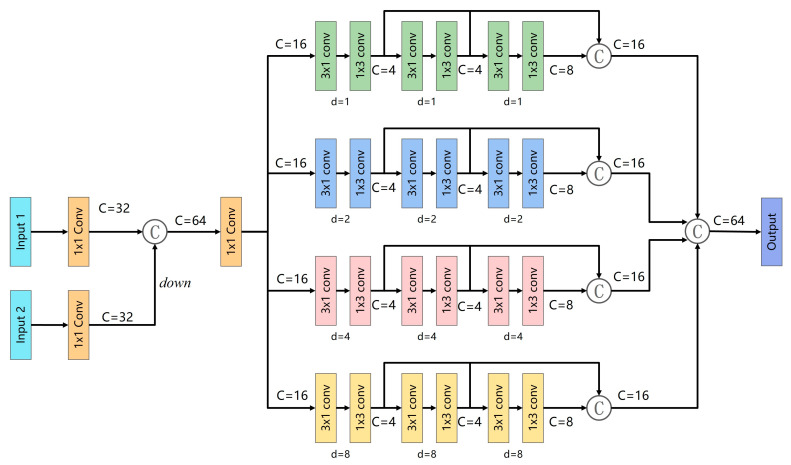
The structure of MCA. It integrates the feature information of adjacent high and low layers and adopts four parallel dilated convolution branches to extract multi-scale features.

**Figure 5 diagnostics-13-00896-f005:**
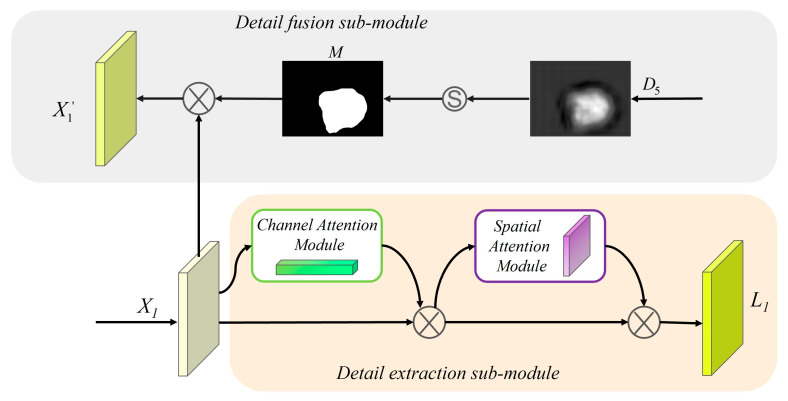
The structure of LDE, which consists of two sub-modules, including the detail fusion sub-module and the detail extraction sub-module. It is used to extract richer low-level details.

**Figure 6 diagnostics-13-00896-f006:**
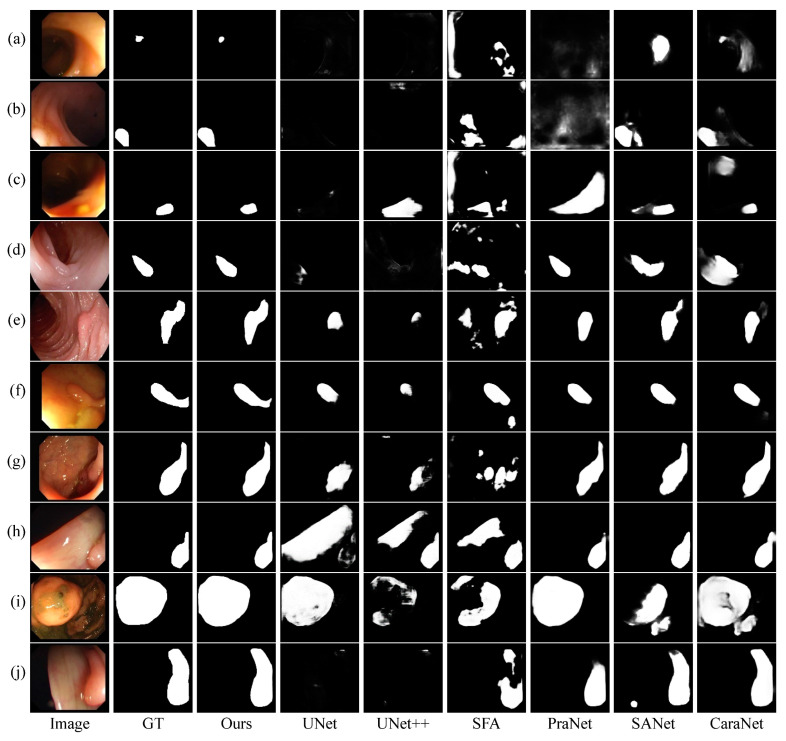
Visualization comparison between the proposed method and six state-of-the-art ones.

**Figure 7 diagnostics-13-00896-f007:**
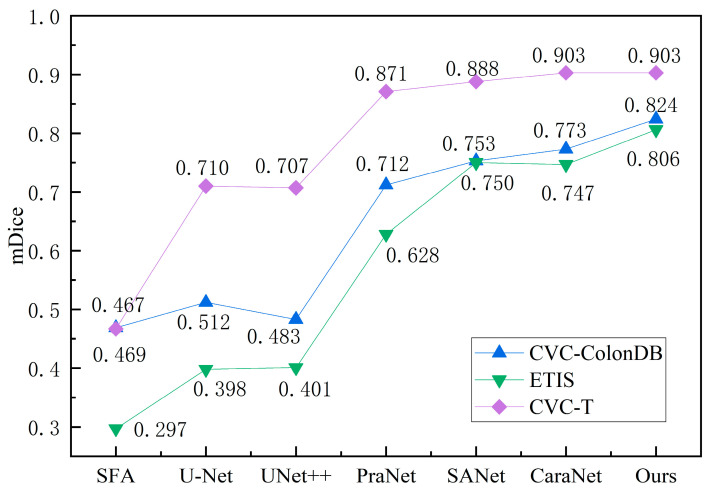
Evaluation of model generalization ability. We provide the mDice results on CVC-ColonDB, ETIS, and CVC-T.

**Figure 8 diagnostics-13-00896-f008:**
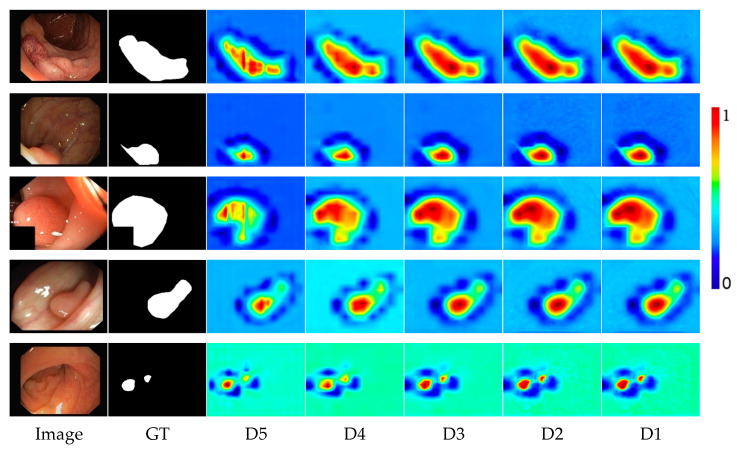
Visualization of output features of the DBE module at different stages. The redder the region, the more attention the network needs to pay.

**Figure 9 diagnostics-13-00896-f009:**
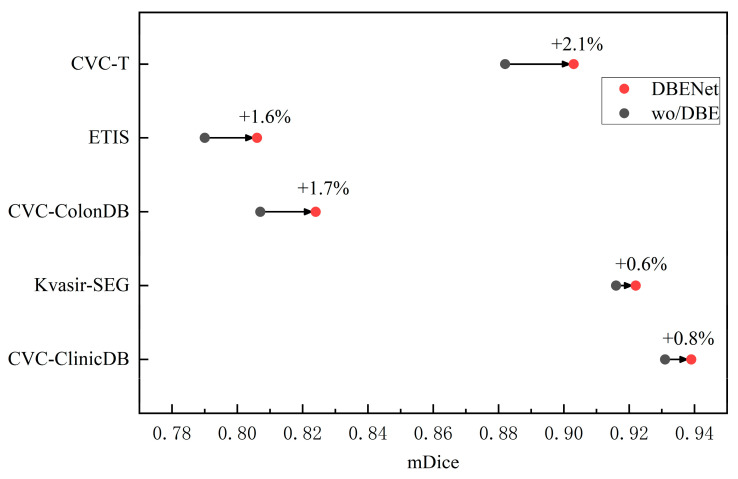
Comparison of mDice results between standard model DBE-Net and “wo/DBE” lacking the DBE module.

**Table 1 diagnostics-13-00896-t001:** The colorectal polyp datasets used in our experiments.

Dataset	Image Size	Image Number	Number of Train Samples	Number of Test Samples
Kvasir-SEG	Variable	1000	900	100
CVC-ClinicDB	384 × 288	612	550	62
CVC-ColonDB	574 × 500	380	0	380
ETIS	1225 × 966	196	0	196
CVC-T	574 × 500	60	0	60

**Table 2 diagnostics-13-00896-t002:** Quantitative comparison of the learning abilities on CVC-ClinicDB and Kvasir-SEG. The best results are shown in bold.

Method	CVC-ClinicDB			Kvasir-SEG		
	mDice	mIou	MAE	mDice	mIou	MAE
U-Net	0.823	0.755	0.019	0.818	0.746	0.055
UNet++	0.794	0.729	0.022	0.821	0.743	0.048
SFA	0.700	0.607	0.042	0.723	0.611	0.075
PraNet	0.899	0.849	0.009	0.898	0.840	0.030
SANet	0.916	0.859	0.012	0.904	0.847	0.028
CaraNet	0.936	0.887	0.007	0.918	0.865	0.023
Ours	**0.939**	**0.895**	**0.006**	**0.922**	**0.871**	**0.022**

**Table 3 diagnostics-13-00896-t003:** Quantitative comparison of the generalization abilities on CVC-ColonDB, ETIS, and CVC-T. The best results are shown in bold.

Method	CVC-ColonDB			ETIS			CVC-T		
	mDice	mIou	MAE	mDice	mIou	MAE	mDice	mIou	MAE
U-Net	0.512	0.444	0.061	0.398	0.335	0.036	0.710	0.627	0.022
UNet++	0.483	0.410	0.064	0.401	0.344	0.035	0.707	0.624	0.018
SFA	0.469	0.347	0.094	0.297	0.217	0.109	0.467	0.329	0.065
PraNet	0.712	0.640	0.045	0.628	0.567	0.031	0.871	0.797	0.010
SANet	0.753	0.670	0.043	0.750	0.654	0.015	0.888	0.815	0.008
CaraNet	0.773	0.689	0.042	0.747	0.672	0.017	**0.903**	**0.838**	0.007
Ours	**0.824**	**0.743**	**0.027**	**0.806**	**0.725**	**0.015**	**0.903**	0.837	**0.006**

**Table 4 diagnostics-13-00896-t004:** Quantitative results of ablation studies of our method under different configurations.

Dataset	CVC-ClinicDB		Kvasir-SEG		CVC-ColonDB		ETIS		CVC-T	
Metric	mDice	mIou	mDice	mIou	mDice	mIou	mDice	mIou	mDice	mIou
Baseline	0.929	0.879	0.913	0.858	0.798	0.716	0.792	0.709	0.891	0.820
w/o MCA	**0.939**	0.892	**0.932**	**0.885**	0.814	0.732	0.801	0.723	0.903	0.836
w/o LDE	0.934	0.885	0.918	0.867	0.818	0.735	0.805	0.723	0.893	0.825
w/o DBE	0.931	0.883	0.916	0.865	0.807	0.726	0.790	0.704	0.882	0.812
Baseline + MCA + LDE + DBE	**0.939**	**0.895**	0.922	0.871	**0.824**	**0.743**	**0.806**	**0.725**	**0.903**	**0.837**

## Data Availability

The datasets used in this research are all benchmark datasets and are publicly available for researchers.

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
