# Peer review of "DBE-Net: Dual Boundary-Guided Attention Exploration Network for Polyp Segmentation"

_diagnostics, 2023, doi:10.3390/diagnostics13050896_

Round 1
Reviewer 1 Report
The paper entitled „DBE-Net: Dual boundary-guided attention exploration network 2 for polyp segmentation” by Ma et al. deals with a dual boundary-guided attention exploration network for polyp segmentation in three stages (i) firstly is solved the boundary-blurring problem; (ii) secondly a multi-scale context aggregation enhancement module is introduced; (iii) low-level detail enhancement module is proposed. The proposed method is tested on five datasets leading to mDice of 82.4% and 80.6%.
The paper presents a coarse-to-fine strategy to approximate the real polyp boundary progressively. To solve the blurred polyp boundaries DBE-Net is proposed. Besides the commonly used methods for segmentation, the authors added the stage for the enhancement of the image.
A few observations:
1.In the BM sub-module, the authors used dilate and erode morphological method, please add the shape of the used structural element.
2. In eq. (9) please add the structural element as the second argument of the dilate and erode methods.
3. In the section Discussions please compare your results with another reference from the scientific literature.
4. In Table 1. For the colorectal polyp datasets used in our experiments, please explain why the number of train samples for CVC-ColonDB, ETIS, and CVC-T is zero.
5. Please add the limitation of the presented study.
6. Please add in the conclusion section the future research.
Reviewer 2 Report
An article written responsibly with multiple data and important details that reveal the need for new techniques. The data presented in the article are publishable, but I can say that a large part of the information presented in the article was difficult for me to process and understand because the article is not written by doctors and is not for doctors. However, I believe that it can be accepted considering that it can be a challenge for the medical community to read this article.
